# Benchmark for Temporal, Ambiguous, and Grounded Embodied Question-Answering

## Abstract

The problem of question ambiguity, while highlighted as an open issue, is often overlooked in the literature on Embodied Question Answering (EQA) and Episodic Memory Question Answering (EM-EQA). This paper proposes a structured approach to handle ambiguity in the egocentric data. Our benchmark, called TAG-EQA, utilizes spatial and temporal grounding to distinguish between objects, positions, and events and ensures that obtained structured answers preserve information fully while effectively resolving ambiguity. We introduce a new dataset, specifically designed for ambiguous grounded Episodic Memory QA. The dataset incorporates situated spatial reasoning, temporal conditions, and diverse visual features. Our new evaluation procedure tackles grounded natural language answers. It reveals that some of the most modern approaches still struggle with efficient information extraction and processing in ambiguous scenarios. We hope that TAG-EQA will serve as both a valuable tool for generating complex EM-EQA data and that the proposed evaluation benchmark will propel progress in agentic AI and embodied reasoning.

## 1 Introduction

A deep understanding of the environment is a prerequisite for successful planning and task execution for agents and robots. One way to measure the ability to reason about surroundings is through the task of *Embodied Question Answering* (EQA). In that setting, the agent is either provided with an observation sequence (passive EQA, or Episodic-Memory EQA (Datta et al., 2022)) or placed in a 3D environment (active EQA (Das et al., 2018)) with the ability to navigate it, and asked a question about the environment. (EM-) EQA is inherently multi-modal and often requires comprehension of spatial, visual and temporal components.

A realistic aspect of EM-EQA, which is often sidelined in the literature, is the potential *ambiguity of the questions*. For example, when asking "How many spoons are there on the countertop" we may have different answers when there exist multiple countertops present in the kitchen scene (see Fig. 1 for an example). Such questions are usually avoided during dataset generation in favor of obtaining an easily verifiable and definite answer, as it keeps the evaluation simpler (Das et al., 2018; Yu et al., 2019; Cangea et al., 2019; Ma et al.; Zhao et al., 2022).

A related issue is *conditioning with 3D grounding* in the dataset. The exact 3D coordinates of the mentioned object allow for questions and task disambiguation, and are useful for further navigation and manipulation (Singh et al., 2023; Gu et al., 2024). Additionally, providing the coordinates proves that the agent truly understands the environment and lowers the probability of learning spurious correlations. This feature is usually present in the datasets devoted to 3D Visual Question Answering (VQA), but is quite frequently absent from EQA.

A further realistic assumption is that *the agent actually changes the scene* in the provided experience. This factor provides an opportunity to test the understanding of the temporal component of the agent's experience by asking questions like "Where was the green cup before you entered the living room?". While agentic actions can be present in the related task of *VideoQA*, the questions there usually target action recognition or very simple temporal aspects of the video (e.g. predicates like "where you seen it last time"), and do not take into account the potential scene changes, induced by the agent.

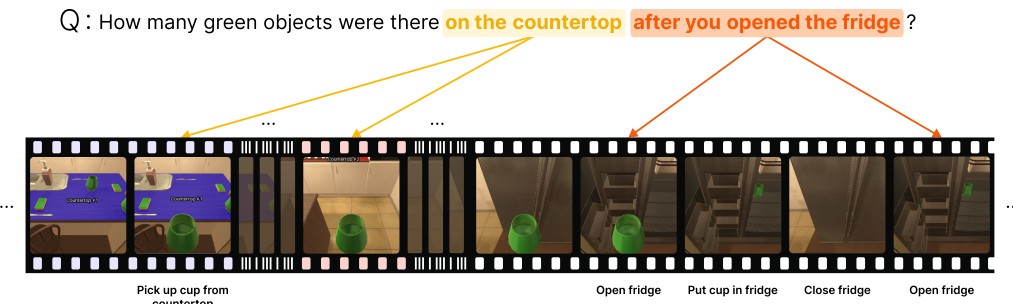

Q : How many green objects were there **on the countertop** **after you opened the fridge** ?

... ...

Pick up cup from countertop

Open fridge   Put cup in fridge   Close fridge   Open fridge

A : On the **countertop** (x=0.46, y=0.15, z=-0.83), there were 3 green objects: a cup (x=-0.43, y=0.21, z=-0.51), and two sponges (x=0.35, y=0.21, z=-0.57), (x=-0.43, y=0.21, z=-0.83), and on the other **countertop** (x=-0.3, y=0.15, z=-2.8), there were no green objects.

Figure 1: EQA agent in open worlds must be able to handle composite questions with diverse conditions, e.g. temporal predicates that indicate a dynamic environment that changes due to the agent's actions, and that might have ambiguity, and provide answers conditioned with 3D grounding to resolve it.

In this work, we present a novel benchmark for Episodic-Memory EQA evaluation called TAG-EQA, which is aimed at simultaneous testing of different aspects of the agentic experience. Through a structured approach, we combine spatial, visual and temporal conditions in the questions and provide spatially grounded answers, and are able to handle ambiguities in generating, answering, and evaluating questions.

Our contributions are as follows:

- a structured approach to ambiguous embodied question answering;
- an interpretable evaluation method for grounded QA in natural language;
- a new dataset that requires advanced spatial, temporal and visual reasoning capabilities.
- an analysis that shows that modern embodied QA methods have trouble answering sophisticated questions in 3D worlds.

## 2 RELATED WORK

Embodied Question Answering (EQA) was first introduced by Das et al. (2018) as an "active setting" alternative to the passive VideoQA. An agent was placed in a simulated 3D household environment, AI2-THOR (Kolve et al., 2017), and was asked a question that was generated from nine possible predefined templates. The setting was active in that the agent had to navigate the environment and gather information to be able to answer correctly. Yu et al. (2019) increased the complexity of the original dataset by the possibility of multiple target objects in the questions. Ever since, many extensions have been proposed. Tan et al. (2023) introduced object functions like "used to cut food" as conditions in the questions. Islam et al. (2023) added multi-modality in terms of verbal and nonverbal gestures. Dorbala et al. (2024) proposed situational queries that require reasoning about multiple objects, and Gordon et al. (2018) introduced a requirement that agents should not only navigate but also interact with objects to answer questions. Despite the ever-increasing sophistication in the types of questions, the active nature of Embodied Question Answering relying on predefined templates makes it difficult to ask questions with a temporal footprint, such as "where was the green mug before you picked it up and placed it on the coffee table?", as no preceding history is usually provided to the agent.

Table 1: Comparison of the unique features of TAG-EQA.

| Dataset | Temporal QA | Grounded QA | Multi-factor QA | Ambiguity |
|---------|:-----------:|:-----------:|:---------------:|:---------:|
| Active EQA | ✗ | ✗ | ✓ | ✗ |
| EM-EQA | ✓ | ✓ | ✗ | ✗ |
| 3D VQA | ✗ | ✓ | ✗ | ✗ |
| **TAG-EQA** | ✓ | ✓ | ✓ | ✓ |

While the active setting of EQA benchmarks is valuable and interesting, in practice, the agent performing tasks in the environment must first determine that it lacks the necessary information to answer the question. This is why the task of *Episodic Memory EQA* (EM-EQA) precedes Active EQA. A dataset exemplifying this is VideoNavQA (Cangea et al., 2019), where the input to the model consists of an agent's trajectory and a question from one of eight predefined categories. Datta et al. (2022) introduced a spatio-temporal localization task, requiring the agent to locate an object based on temporal conditions, such as "seen for the first time" or "seen last time." Similarly, EgoVQ3D (Grauman et al., 2022) tasks the agent with finding the last visible position of an object. QaEgo4D (Bärmann & Waibel, 2022) does not require precise localization and accepts natural language answers, as does the EM-EQA subset of OpenEQA (Majumdar et al., 2024). However, these works generally involve relatively simple temporal conditions (e.g. first or last time), and natural language variations do not provide 3D coordinates as grounding in their QA pairs, making them vulnerable to guessing.

For understanding nuanced spatial relationships, 3D VQA datasets like ScanQA (Azuma et al., 2022) or FE-3DGQA (Zhao et al., 2022) are more appropriate. These datasets typically rely on realistic 3D reconstructions of environments and use these data as input (e.g., ScanNet, Matterport 3D (Dai et al., 2017; Chang et al., 2017)). SQA3D (Ma et al.) introduces situated embodied reasoning, where the agent's position is described textually (e.g., "sitting on the edge of the bed and facing the table") and is used as a condition in the question (e.g., "Which direction should I go to heat my lunch?"). Although 3D VQA datasets can incorporate not just top-down views or complete reconstructions but also provide as input sequences of observations, grounding, and narrow down the sim-to-real visual gap, they still suffer from the same limitations as other realistic datasets, such as OpenEQA (Majumdar et al., 2024) and Wijmans et al. (2019). Specifically, they lack agent-environment interactions, which limits the diversity of temporal conditions that can be explored.

Although Embodied Question Answering (EQA) has clearly been studied extensively in the literature, it still suffers from key limitations in the complexity of questions it admits. An obvious limitation is the explicit avoidance of ambiguity in questions (Das et al., 2018; Yu et al., 2019; Cangea et al., 2019; Ma et al.; Zhao et al., 2022). For instance, a question like "Where did you leave the book after you went to the living room?" might have multiple possible answers if the agent was carrying multiple books. This limitation may be a forced measure, but nevertheless an unrealistic assumption. The grounding in the form of 3D coordinates of objects, which would resolve spatial ambiguity and provide useful information for further planning, is also not provided in the EM-EQA setting. Another limitation, which was mentioned earlier, is the lack of temporal conditioning in the questions due to the small diversity of events (i.e. EM-EQA), or inherent inability for incorporating such conditions (i.e. Active EQA, 3D VQA).

Our benchmark that adds support to Embodied Question Answering for all the aforementioned types of conditions (i.e. spatial, temporal, visual, and functional) and combinations thereof, as we also summarize in Table 1.

## 3 TEMPORAL, AMBIGIOUS, AND GROUNDED EMBODIED Q&A

Here, we present Temporal, Ambigious, and Grounded Embodied Q&A (TAG-EQA), an EM-EQA benchmark designed to evaluate question answering in natural language across diverse spatial, temporal, and visual dimensions, with a focus on grounding and ambiguity resolution. To generate the

questions, we use ground truth data from the environment, including the positions and states of both objects and the agent. The agent's trajectory – comprising a sequence of actions, observations, and position recordings at each timestep – serves as the primary input. Each question is constructed in a structured, functional manner, following the original EQA approach (Das et al., 2018):

$$\text{DATA} \rightarrow \text{FILTER}_1() \rightarrow \cdots \rightarrow \text{FILTER}_n \rightarrow \text{QUERY}()$$

We use a set of filtering functions, each representing a specific predicate for the objects (e.g., properties like "color = green", or spatial relations like "location: on the object = countertop"), spatial situated predicates (e.g., "where = a chair is in front of a dining table", "where = two green cups are in front of you"), and temporal context (e.g., "when = before you picked up a cup"). Each predicate consists of a "key" and a "value." These predicates are combined using an AND operator to form the conditions of the question. The data is then filtered through the corresponding filtering functions, and a random query function (e.g., "How many . . .", "Where . . .") is applied to generate the question. Unlike most (EM-)EQA datasets, we do not apply the "unique()" operator at the end, meaning the answer to the question may remain ambiguous. For a complete list of predicates and queries, refer to Table 3 and Table 4 respectively.

**Ambiguity.** In our benchmark, ambiguity arises when two distinct objects, agent positions, or moments in time satisfy the same predicate key but have different values. For example, two spoons on countertop #1 and countertop #2 both satisfy the predicate "location: on the object = countertop", but each has a different value for the predicate. To handle such questions and provide an answer, we employ a structured approach, grouping answers by the specific predicate values they satisfy. This ensures that the ambiguity in the question is resolved and the answer is complete, and no information has been lost. For an example, refer to Fig. 2.

**Grounding.** To obtain such a structure, where the predicate values are distinct (e.g. "on the countertop #1" and "on the countertop #2"), it is necessary to differentiate between objects of the same type by assigning tags. Since we have access to the ground truth state of the environment, we use the 3D coordinates of objects, captured at the correct moment, as their tags. The 3D coordinate system is chosen such that the agent is placed at the origin at the time of the first observation. For each ambiguous question, we provide a grounded counterpart by referencing the coordinates of each object

```
{
    "structured_question":"countObjects? (property:color = green) AND (
    location:on the object = countertop) AND (when = after open fridge)",
    "structured_answer":    [
        {
            "predicate":"location:on the object = countertop (x=0.46, y
    =0.15, z=-0.83)",
            "result":[
                "cup (x=-0.43, y=0.21, z=-0.51)",
                "sponge (x=0.35, y=0.21, z=-0.57)",
                "(x=-0.43, y=0.21, z=-0.83)"
            ],
            "value": 3
        },
        {
            "predicate":"location: on the object = countertop (x=-0.3, y
    =0.15, z=-2.8)",
            "result":[],
            "value": 0
        }
    ]
}
```

Figure 2: An example of a structured question-answer pair in the JSON format, that corresponds to the ambiguous question asked in natural language in Fig. 1.

in the question (e.g., "How many spoons are there on the countertop (x=-0.1, y=1.1, z=0.5)?"). This is similar to pointing at the object, effectively resolving any potential spatial ambiguity.

**Multiple factors.** Our framework includes a rich set of predicates, such as diverse visual and functional properties, temporal conditions involving various events, and both situated and relative spatial reasoning. To combine these predicates and generate meaningful questions, we employ several strategies. First, we assess whether each predicate condition in the question is *effective*, meaning that adding the condition alters the final answer. This allows us to avoid a common sense conditions like "cups that are used for drinking". Additionally, as the final answer changes with every new condition, the effectiveness property allows to detect if the model is ignoring parts of the question. The same principle is applied to grounded questions; for example, if there are only green objects on the dining table, the question "What green objects are located on the dining table (x=1.1, y=0.8, z=-2.3)?" is not considered effective. Second, we measure the diversity of predicates to prevent questions in which the majority of conditions are of the same type. This ensures the model works with a mix of spatial, temporal, visual, and other data, increasing the complexity of the question. Lastly, to avoid giving unintended hints to the model, we restrict some predicates from appearing alongside specific query functions, such as "Where are the spoons located on the countertop?", since "countertop" would also be part of the answer.

**Natural Language Question-Answer Generation.** After generating question-answer pairs in structured form, we convert them into natural language by prompting a Large Language Model (LLM). To achieve this, we provide the LLM with several examples from the training portion of the dataset. The goal is to produce natural-sounding and diverse question formulations while preserving all essential information.

**Fine-grained evaluation.** Since our generation process is controllable, allowing us to track the predicates used, we can control the stratification in the train-test split. Moreover, with fine-grained evaluation, we can now identify specific conditions or combinations of conditions where the model struggles. The synthetic nature of the dataset provides ground truth object semantic segmentation, object matching, and access to precise visual properties. This allows for the isolated testing of specific components of a composite model, such as evaluating the model's retrieval abilities without introducing errors from object recognition.

**Dataset statistics** We use FILM (Min et al., 2021) as the agent model, and the test subset ALFRED dataset (Shridhar et al., 2020) of everyday household tasks in AI2-THOR (Kolve et al., 2017) virtual environments. The choice of the test part of ALFRED is determined by the desire to obtain realistic trajectories, in which the agent can be looping in some place, and perform unsuccessful actions. The utility properties of each object were extracted using prompted GPT-4o (Achiam et al., 2023). In total, we generated around 1.5k trajectories, with a maximum length of 250 time steps. Using these trajectories as data, we produced a total of 14k QA pairs, 70% of which we reserved for train part, and the rest was split into validation and test. The stratification strategy was chosen such that half of the validation and test sets have questions from environments that are not present in the train part. To avoid skewing into one question type, we perform further stratification by the generation parameters, e.g. predicate and query types, and presence of grounding. We used the Gemma-2 27B tuned instruction (Team et al., 2024) as our LLM for the generation and evaluation of natural language QA.

## 3.1 Code-Judge LLM Evaluation

To evaluate the answers in natural language, we employ LLM-as-a-judge (Zheng et al., 2023), similar to the OpenEQA. However, the OpenEQA method provides no reasoning behind the score, it does not take into account factors like the proximity of the identified objects to their true positions, and it relies only on the agreement between the LLM score and the human judgement, which is not suitable for our grounded and structured question-answer generation. We, instead, prompt the LLM to provide *the scoring function* for the given sample, which shall take into account the 3D positions of the objects, their relative locations, their type, visual features, etc.

Our Code-Judge LLM generates a scoring function in Python by applying the same structured approach, i.e. extracting the structure of the scene from the answer and matching the objects. For both

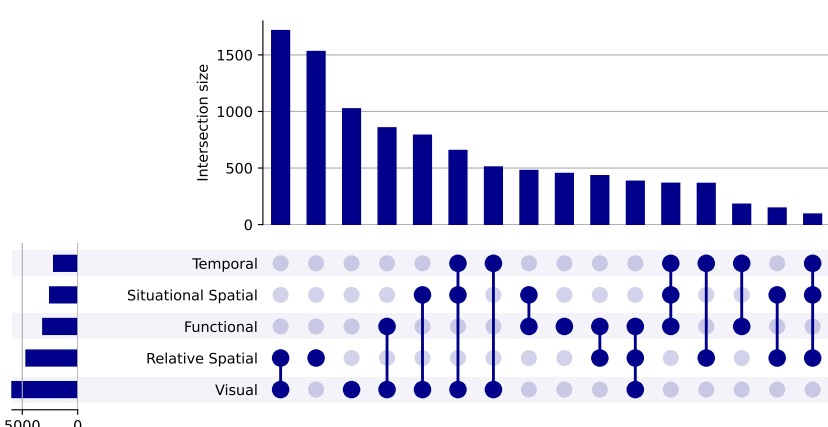

Figure 3: The distribution of different types of conditions in the dataset. "Situational spatial" refer to the "where" predicates, where the agent's position is described, and "Relative spatial" refer to "location" predicate, describing the positions of the objects relative to each other.

correct and generated answer, it extracts the objects tags, positions and captions, and groups them by the predicates they satisfy, e.g. "location = on the countertop (x=.., y=.., z=...)" . The groups of predicates are then matched using the semantic similarity and positional information, provided in the predicate. Afterwards, the objects that belong to the matched predicates are matched between each other using a Hungarian algorithm. The cost matrix for the matching is combined from semantic similarity of object tags, their captions and the positional information, by summing them with some weights. The final matching score is obtained by summing all the matching scores between the objects, weighted by corresponding predicate matching score, and normalized and projected into the 1-5 interval.

**Metrics.** Our scoring implementation provides two metrics: one with grounding, i.e. taking into account the provided coordinates in the questions, and the other one ignores this info. The positional similarity is obtained from the distance $d$ between the objects and normalized with the following formula:

$$\text{possim}(d) = \exp\left(-\frac{d^2}{2\sigma^2}\right)$$

where $\sigma = 1.65$ is tuned so that the score is close to 1.0 within one meter radius, and is close to 0.0 around 5 meters distance. The semantic similarity score is calculated using a Sentence-T5 encoder (Ni et al., 2022). For an example of a scoring program, please refer to Fig. 7.

**Validating the scoring method.** To validate that the proposed scoring method is effective, we perform the following experiment. Using the subset of 300 samples from the validation set, we perturb the correct structured answers applying three randomly chosen disruptions from the following list: random object tags substitution, perturbation in positions, features and predicates. We calculate the matching score between true and perturbed answers with a hard-coded solution, and arrive at the score of 3.5. We then convert perturbed answers into natural language with the same LLM, and calculate the score using our Code-Judge LLM. We observe a high Pearson correlation between two scores of 0.81.

## 4    BASELINES

We run a number of baselines that are capable of generating a natural language output. We adapt them to handle temporal component (e.g. manipulative actions) and grounding in our dataset.

**Socratic LLM + Image captions.**   Following Majumdar et al. (2024), we run Socratic LLM (Zeng et al., 2022) with image captions (Socratic LLM+IC). We prompt the model with the history of actions and observations, where each action consists of a natural language description and an indicator of success, and each observation consists of a caption and an agents position. To reduce the redundancy of the input information and to satisfy the LLMs context length limit, we subsample up to K=50 observations. We make sure that the question is answerable from the provided information, by keeping the subset of important observations such that every manipulation action like "opened the fridge" or "picked up apple" is picked and each object is visible at least once. For this, we treat each manipulation action event and each object as a node in a graph, and each observation as an edge. The nodes have an edge between them if they are captured in the same observation. We extract the minimum spanning tree of this structure to obtain the final list of essential observations. We use instruction-tuned Gemma 2 with 9B parameters (Team et al., 2024) as a base LLM, and employ CogVLM 2 (Hong et al., 2024) for image captioning as our Vision-Language Model (VLM) here and in further baselines.

**Socratic LLM + Image captions + Object recognition.**   As the obtained image captions do not include the 3D positions of objects, outputting grounded answers by the Socratic LLM + Image captions baseline is almost impossible. Besides, the produced image captions may not include the information on the valuable objects in the image and their relationships. To tackle this limitation, we enhance the previous baseline with object recognition information. We add the visible objects tags to the VLM prompt to obtain object-focused image captions, and add a list of objects with their estimated 3D positions to the observation description. We use ground truth objects masks, object matching, and depth estimation, obtained from the simulator.

**LLM + ConceptGraphs.**   ConceptGraphs (Gu et al., 2024) is a structured method for representing scenes, capturing objects' 3D positions, their relationships, and descriptive captions. We adapt this method for our EM-EQA setting. We generate captions for objects with our VLM and then summarize them with the LLM. We assume there is only one acting agent in the scene and first prompt the Large Language Model (LLM) with the history of actions and observations. Each observation here consists of a list of observed object IDs obtained from the ConceptGraph. For manipulation actions, we also include a short description of the target object, such as its last recorded 3D bounding box and center, and ID. Finally, we provide a complete list of objects from the ConceptGraph, along with their relationships. This approach enables us to capture both the dynamics of the scene and the accumulated knowledge of object positions.

**LLM + API.**   Even structured approaches like LLM + ConceptGraph can suffer from information redundancy and may exceed the context length limit of the LLM. To address this, we propose a new baseline based on Retrieval-Augmented Generation (Lewis et al., 2020), built on top of the ConceptGraph approach. This baseline splits the generation process into two steps.

First, we provide the LLM with a retrieval API consisting of three functions:

- `filter_by_semantic_similarity(observations, query, ...)`, which filters observations based on the similarity between the provided natural language query and either the observation caption, object caption, or action description.

- `filter_by_position(observations, position, ...)`, which retrieves positions within a specified threshold. In our case, we set the threshold to one step length, i.e., 0.25m.

- `are_semantically_similar(text1, text2)`, which computes the semantic similarity between two text entries, used to check object properties such as utility based on the object tag.

For semantic similarity estimation, we use the Sentence-T5 large model (Ni et al., 2022) and compute cosine similarity between the text embeddings. Additionally, we include the question in the retrieval prompt to ensure query-specific responses. In the second step, the retrieved relevant observations, along with the query, are provided to the LLM, which is prompted to generate the final answer.

The complete API description is provided in Fig. 8.

Table 2: Evaluation results of the baselines.

| Model | Score | Grounded score |
|---|---|---|
| Socratic LLM + Image captions | 1.33 | 1.29 |
| Socratic LLM + Image captions + OR | 1.62 | 1.53 |
| LLM + ConceptGraphs | 1.54 | 1.46 |
| LLM + API | **1.80** | **1.70** |

## 5 RESULTS

**Evaluating Temporal, Ambiguous, and Grounded Q&A.** We provide the overall accuraces on the benchmark for the various baselines in Table 2. In general, the specific object-centric captions are more useful and increase the score. That said, the LLM+API scores the highest, and the LLM+ConceptGraphs achieves almost the same score as the Socratic LLM + IC + OR. Thus, it can be said that the amount of relevant, query-specific information, provided to the model, plays a huge role.

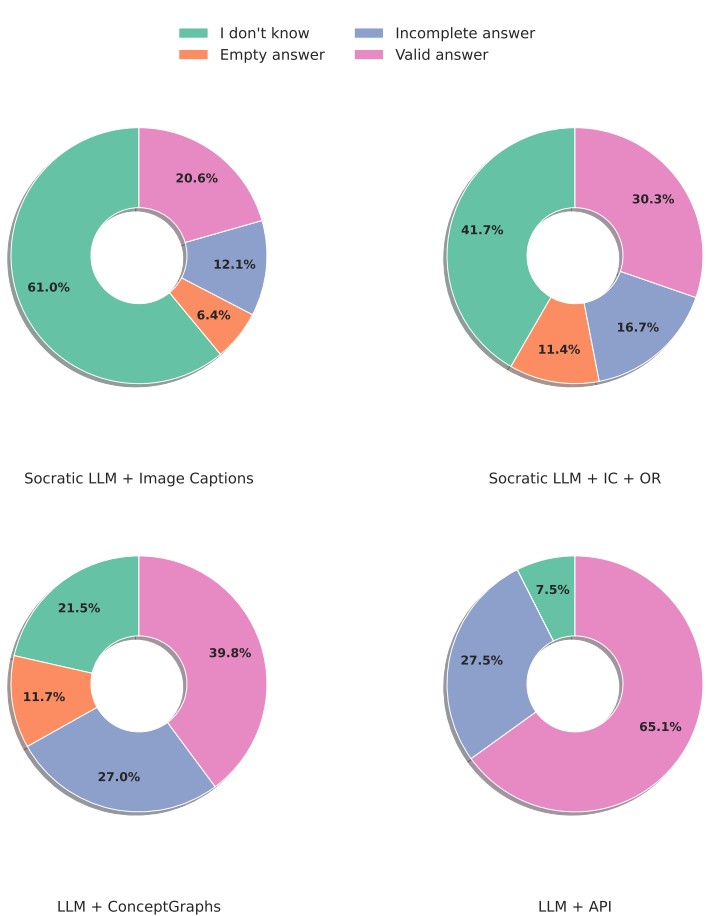

Figure 4: The distribution of exceptions encountered in the answers. Valid answers are answers that are not empty, so they are not necessarily correct. The percentage of valid answers, i.e. that are complete and provide some sort of information, is increasing with structure and relevancy of the input.

As expected, as the grounding score takes into the account the positions of the mentioned objects, the score is always lower. Given that all baselines seem to be scoring much lower than the perfect 5, we conclude that current embodied language models have hard time understanding the spatiotemporal extent of the environment and of the questions.

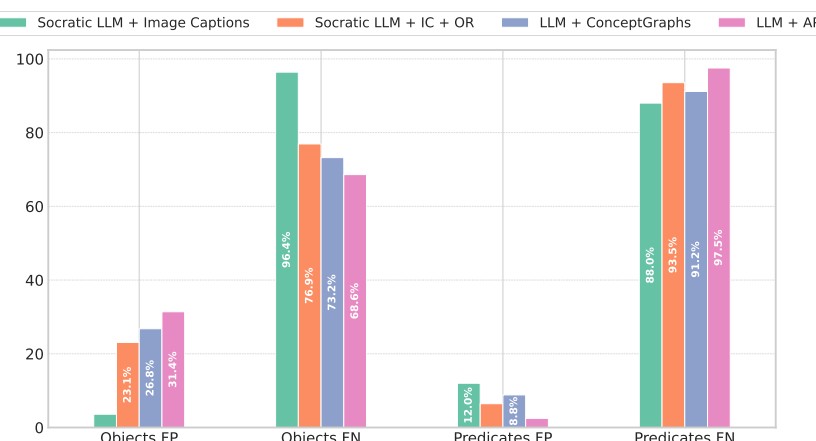

Figure 5: The distribution of the error types.

**Structure helps providing valid answers.** We dive deeper into the performance of the baselines, first analyzing the distribution of validity in the answers in Fig. 4. Valid answers are answers that are not *a priori* incorrect, e.g., because they are empty. As the overall percentage of invalid answers drops with both (a) adding more object-centric information (i.e. Socratic LLM + IC + OR) and (b) adding structure and retrieval (i.e. LLM+API), we conclude that the they are both valuable, with the following analogy: the first one for the "recall" part, i.e. making sure that the valuable for the answer information is in the input, and the second for the "precision", i.e. retrieval of this relevant information and its efficient processing.

**Strong baselines struggle with even simple predicates.** In Fig. 6 we analyze in further depth the accuracies of the different baselines per category of question. While there are some fluctuations, it seems all baselines struggle equally as much in all categories of predicates: situational and relative spatial (e.g., "where = the couch is in behind the cabinet", "location = in the fridge"), visual and functional (e.g., "color = green", "purpose = used for cutting food"), temporal (e.g. "when = right before you lit the candle") and so on. We notice that the object-targeted captions in Socratic LLM + IC + OR help with relative spatial relations, probably by capturing these relationships with the VLM, but still struggle with situational spatial questions, that require advanced understanding of the geometry of the provided experience (e.g. "I turned right, now the previous observation is on my left".). The accurate retrieval helps the most with both kinds of spatial question categories. Quite naturally, adding more specific visual information processing in the Socratic LLM + IC + OR is important for successful visual and functional question answering. As all models essentially struggle even with simple one-condition questions to the point close to failure, adding multiple conditions does not have much of an effect. Thus, we conclude that the LLM-based baselines do not have the required tools or capabilities to reason about various factors of 3D environment.

**Distribution of types of errors.** An interesting pattern can be noticed in the Fig. 5. As we add more structure to the input, although the model tends to include more irrelevant objects in the answer (i.e. increase in "Objects False Positive"), it also more frequently retrieves the relevant objects (i.e. reduced "Objects False Negatives"). Thus the structure and retrieval increases the coverage in the objects, but reduces the precision. For the predicates, the picture is unexpectedly inverted: more structure means less hallucinations in the predicates (fewer false positives), but the correct predicates may be missing from the answer, which means that the model is struggling to resolve the ambiguity using multiple predicates.

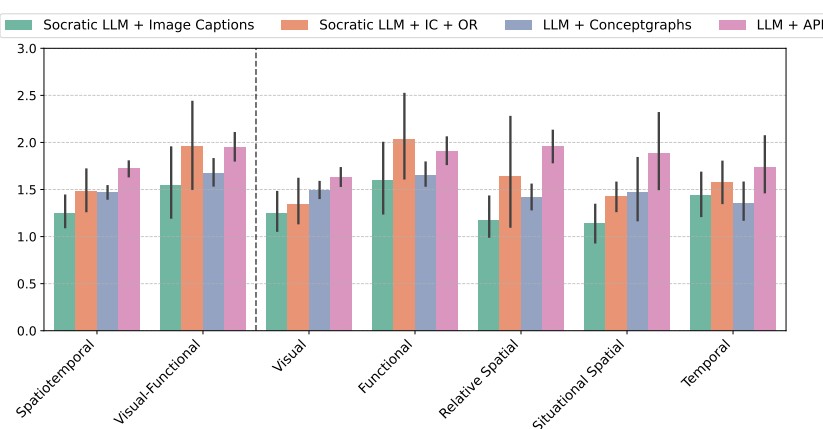

Figure 6: The distribution of scores per condition category and categories of questions with combined conditions (Spatiotemporal and Visual-Functional).

## 6 CONCLUSION

**Limitations.** Due to the increased complexity of processing and filtration, we avoided the potential temporal ambiguity in our dataset by adding the counter to each event (e.g. "open the fridge for the second time"), thus making it unique. Though the temporal predicates are still diverse, we think that handling this ambiguity could be a straightforward extension of our dataset which doesn't require any modifications to the data generation procedure.

While incorporation of temporal component into the questions of Active EQA is not obvious, we can propose a new *Mixed EQA*, a harder task, where the agent is provided with both preceding trajectory and is placed in the environment. This task is the most challenging and realistic, and would allow us to use temporal predicates to ask questions like "What objects were to the right of you when you placed the cup on the table?".

**Reproducibility Statement.** To ensure the reproducibility of our results, we aim to make the TAG-EQA dataset and evaluation procedures publicly available, along with detailed documentation on data preprocessing, generation, LLM prompts and code for baselines. All model architectures and parameters used in our experiments are clearly specified in the main text. We encourage the community to build upon our work by evaluating their models and generating more sophisticated versions of the benchmark.

In this paper, we presented TAG-EQA, a benchmark designed to address the issue of complex, ambiguous, and multi-factor questions in Episodic Memory Question Answering (EM-EQA). By leveraging structure and spatial grounding, our approach effectively provides answers that resolve ambiguities related to objects, positions, and events, ensuring the preservation of essential information. Through a novel dataset and evaluation procedure, we demonstrated that while structured input data and retrieval techniques improve performance in spatial reasoning tasks, challenges remain with even simple one-condition queries. Our results show that current strong methods continue to struggle in ambiguous and grounded scenarios, underscoring the need for further advancements in embodied reasoning. We hope that TAG-EQA will foster new research directions in the development of more robust systems for handling complex, multi-factor, ambiguous queries in EQA environments.

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

## A APPENDIX

### A.1 DATASET

### A.2 EVALUATION

### A.3 RETRIEVAL API

Table 3: Complete list of queries. Each modification entry changes the initial query to create a different question.

| Query type | Modifications | Question |
|---|---|---|
| listObjects | | What objects ...? |
| count | object | How many objects ...? |
| | time | How many times ...? |
| getState | open | Is the object ... opened or closed? |
| | toggled | Is the object ... toggled on or off? |
| | broken | Is the object ... broken or intact? |
| | sliced | Is the object ... sliced? |
| | filled with liquid | Is the object ... filled or empty? |
| getProperty | color | What is the color of the object? |
| | utility | What is the object used for? |
| | material | What is the object made of? |
| getLocation | | Where is the object? |

Table 4: Complete list of predicates.

| Predicate type | Modifications | Examples |
|---|---|---|
| Property | color | "color=green" |
| | utility | "utility=used for cutting food" |
| | material | "material=plastic" |
| Location | on / inside the <OBJ> | "inside the fridge" |
| | on / inside same <OBJ1> as <OBJ2> | "on the same coffee table as the apple" |
| | above / under the <OBJ> | "under the dining table" |
| State | isOpen | "microwave is open" |
| | isToggled | "candle is lit  the light is switched on", |
| | isBroken | "statue is broken" |
| | isSliced | "tomato is sliced" |
| | isFilledWithLiquid | "bowl holds liquid" |
| Position | <OBJ1> is in front of / behind <OBJ2> | "the sofa is in front of the TV" |
| | <OBJ1> is to the left / right of <OBJ2> | "coffee machine is to the left of the cup" |
| Moment | before / after <EVENT> | "before you picked up the knife" |
| | right before / after <EVENT > | "right after you opened the fridge" |
| | between <EVENT1> and <EVENT2> | |

```python
# Generated answer: There are 3 cups on the countertop (x=0.1, y=0.23, z
    =-0.3).
def score():
    gt_answer = [
        {
            "predicate": "location = on the countertop (x=...)",
            "objects":[
                ...
            ],
            "value":3
        },
        {
            "predicate": "location = on the countertop (x=...)",
            "objects":[],
            "value":0
        }
    ]
    predicted_answer = [
        {
            "predicate": "location = on the countertop (x=0.1, y=0.23, z
    =-0.3)"
            "objects":[
                {"type": "cup", "position": None, "caption": "green"},
                {"type": "cup", "position": None, "caption": "green"},
                {"type": "cup", "position": None, "caption": "green"}
            ],
            "value":3
        }
    ]
    return calculate_matching_score(gt_answer, predicted_answer)
```

Figure 7: An example of a scoring program produced by our LLM-as-a-judge method, that correspond to the running example from Fig. 1.

```python
def filter_by_semantic_similarity(args) -> List[Dict]:
    """
    Filter observations by semantic similarity between the provided
    caption and the observation, object or action.

    Args:
        observations (List[Dict]): List of observations.
        type (Literal['observation_caption', 'object_caption', 'action'])
    : Type of the caption to filter by.
        caption (str): Caption to filter by.
        threshold (float): Threshold for the semantic similarity.

    Returns:
        List[Dict]: Filtered observations.
    """
    return filtered_observations

def filter_by_position(args) -> List[Dict]:
    """
    Filter observations by closeness to a position.

    Args:
        observations (List[Dict]): List of observations.
        object_type (str): Type of the object to filter by.
        position (Dict[str, float]): Position to filter by.
        type (Literal['object_position', 'agent_position']): Type of the
    position to filter by.
        threshold (float): Threshold for the position.

    Returns:
        List[Dict]: Filtered observations.
    """
    return filtered_observations
```

Figure 8: The retrieval API definition of the LLM+API baseline.

