# OpenReview forum: "Benchmark for Temporal, Ambiguous, and Grounded Embodied Question-Answering"
_ICLR.cc/2025/Conference — ICLR 2025 Conference Withdrawn Submission_

### Official Review · Reviewer_RCjF · 2024-10-31

**Soundness:** 3
**Presentation:** 3
**Contribution:** 3
**Rating:** 5
**Confidence:** 4

**Summary:**

This paper introduces a new benchmark called TAG-EQA, which focuses on addressing question ambiguity in Embodied Question Answering and Episodic Memory Question Answering. The proposed benchmark utilizes spatial and temporal grounding to differentiate between objects, positions, and events, ensuring ambiguity is resolved with structured answers. To evaluate performance in ambiguous scenarios, the paper presents a new dataset specifically designed to handle ambiguity in episodic memory QA, incorporating spatial reasoning, temporal conditions, and diverse visual features.

**Strengths:**

● This paper focuses on the impact of ambiguity in real-world scenarios on AI methods, which is both interesting and practically valuable. Compared to randomly constructed QA, considering ambiguity makes the problem more challenging. To represent ambiguity, the authors introduce structured answers to prevent information loss.
● The paper also proposes a novel method for synthesizing a large number of challenging and accurate question-answer pairs for embodied reasoning by combining conditions and using a filtering strategy. Compared to widely used data synthesis methods based on multimodal large language models, this approach can provide theoretically sound synthesized results, assuming the original data annotations are accurate.

**Weaknesses:**

● Although the authors demonstrate the diversity of condition combinations in Figure 3 and claim in lines 228-229 that they measure the diversity of predicates to prevent questions where most conditions are of the same type, they do not provide statistical information on the distribution of predicates. The richness of predicates more accurately reflects whether they meet real-world demands. Could the authors provide more evidence to suggest that the predicate set is diverse enough? For example, a breakdown of predicate types and their frequencies that demonstrates the diversity of the predicate set would be helpful. Additionally, more examples of complex questions with multiple conditions would also help illustrate the diversity.

● The authors use LLM combined with various image-to-text methods to perform multimodal tasks. However, as they mention, these approachs can lead to information loss and redundancy. Why the authors consider only evaluate on LLMs? It is also recommended that the authors could provide experimental results based on multimodal large language models, such as the flagship models GPT-4o, Qwen2-VL, and models developed for handling long temporal sequence data, such as LLaVA-OneVision and mPLUG-Owl3, etc. The evaluation results on these models can better reveal the challenges posed by the proposed benchmark for end-to-end multimodal methods.

● The proposed method heavily depends on the accuracy of object properties annotations; incorrect annotations not only create erroneous samples directly but also interfere with the combination and filtering strategies, leading to more cascading errors. However, as stated in line 251, the properties of objects are extracted by GPT-4o. How do the authors ensure the accuracy of the dataset samples? For instance, have the authors performed any manual checks, or employed additional methods to filter out incorrect samples?

**Questions:**

please see the above weakness section.

---

### Official Review · Reviewer_9CJM · 2024-11-04

**Soundness:** 2
**Presentation:** 2
**Contribution:** 2
**Rating:** 5
**Confidence:** 3

**Summary:**

The paper introduces the TAG-EQA benchmark for addressing ambiguity in Embodied Question Answering (EQA) and Episodic Memory Question Answering (EM-EQA) by focusing on egocentric data ambiguity resolution through structured approaches. The benchmark uses spatial and temporal grounding to distinguish between objects, positions, and events, and introduces a new dataset designed specifically for ambiguous grounded episodic memory QA.

**Strengths:**

1. Introduces a novel benchmark (TAG-EQA) that focuses on spatial, visual, and temporal ambiguities in questions
2. Includes a new dataset that incorporates spatial reasoning, temporal conditions, and visual features, making it highly relevant for developing more capable EQA systems.

**Weaknesses:**

1. The benchmark improves temporal reasoning but does not fully address temporal ambiguities, overly simplifying scenarios by using counters for repeated actions.
2. The structured approach and detailed evaluation might be difficult to scale due to the complexity of processing and generating queries that handle multiple ambiguity layers.
3. Experiments are not solid enough.
4. Paper structure should be optimized.

**Questions:**

1. No exact data statistics.
2. There are too few query types.
3. Figure 4 should be optimized.
4. Have you tried to test the performance on other LLMs and VLMs, such as GPT4 and LLaVA?
5. Section 3.1 looks weird in section 3. Can you reorganize this section?

---

### Official Review · Reviewer_dxjv · 2024-11-04

**Soundness:** 1
**Presentation:** 1
**Contribution:** 2
**Rating:** 3
**Confidence:** 4

**Summary:**

This paper presents a benchmark for Episodic-Memory EQA of TAG-EQA, evaluating spatial, temporal, multi-factor and visual reasoning capabilities. Authors fail to locate their experimental settings precisely among existing studies. I am hesitant to say but some sections of this paper seem still incomplete and this paper might be submitted before finishing it.

**Strengths:**

1. Evaluation results of the baselines are supplied.

**Weaknesses:**

1. Confusing terminologies. They do not follow terminologies that are used in the precedent studies. See questions.
2. Ignorance of the branches of the existing VideoQA benchmarks as they introduced so-to-say “passive EQA” and seem to abandon the efforts to position their studies among branches of text-video studies. They discuss the paper contributions without groundings to video QA references.
3. Lack of the task overviews and dataset examples. Because of the confusing terminologies, the overview of the tasks and settings is not well-depicted.
4. Lack of qualitative analyses with images or figures.
5. Conclusion section is not well-organized. Appendix also seems to lack textual descriptions.
6. Lack of the details of the “multi factor” and why they are considerably novel in their dataset compared to existing datasets. L219 discussed their framework includes diverse predicates, without statistics or qualitative comparisons.

**Questions:**

1. Please position your experimental settings as VideoQA (including episodic memory QA) or embodied QA. Otherwise the paper itself seems totally confusing. See the discussion below.
2. L. 32: “passive EQA, or Episodic-Memory EQA (Datta et al., 2022)” This wording at the beginning of the introduction is totally confusing. Although Datta mentioned embodied question answering (EQA) in their paper, they called their study Episodic Memory Question Answering (EMQA) in Datta et al., 2022, not EQA nor even EM-EQA.
3. Can you also explain in which the terminology “passive EQA” was introduced in the existing studies? For example, “Embodied Question Answering” of Das et al., 2018 classified Embodied Question Answering inheriting active actions and VideoQA inheriting passive in their second figure. If “passive EQA” existed, what’d be the difference from VideoQA?
4. What is the form of the proposed dataset? Is Figure 1 an instance of TAG-EQA, although it is not explicitly specified? If so, are there no such questions in the existing QA datasets?
5. L 120-122: “This is why the task of Episodic Memory EQA (EM-EQA) precedes Active EQA.” I cannot understand what the authors meant here. EQA agents might perform the episodic memory QA from their egocentric views AFTER their exploration.
6. L 044-047: “Additionally, providing the coordinates proves that the agent truly understands the environment and lowers the probability of learning spurious correlations. This feature is usually present in the datasets devoted to 3D Visual Question Answering(VQA), but is quite frequently absent from EQA.” It’s not clear what this “spurious correlations” means yet. Can you clarify this further and why this “spurious correlations” exists in 3D-VQA but not in EQA? Are these discussed in the existing studies? If so, can you cite them? Note that this is crucial for defending the novelty of this dataset.
7. As an educational comment, I encourage authors to include much more existing EQA, VideoQA and related embodied or video benchmarks. The current Table 1 is too simplified to access the virtue of this work.
8. Gemma's citation style is strange: Team 2024?

---

### Official Review · Reviewer_tvdV · 2024-11-04

**Soundness:** 2
**Presentation:** 1
**Contribution:** 2
**Rating:** 3
**Confidence:** 4

**Summary:**

The paper presents TAG-EQA, a benchmark for Embodied Question Answering (EQA) that addresses the often-overlooked issue of question ambiguity in existing datasets. The authors propose a structured approach that incorporates spatial and temporal grounding to enhance the accuracy and relevance of answers in complex scenarios. The main contribution lies in the introduction of a new dataset specifically designed for ambiguous grounded Episodic Memory Question Answering, which requires advanced reasoning across spatial, temporal, and visual dimensions.

**Strengths:**

1. The methodology for developing the TAG-EQA benchmark is well thought out.

**Weaknesses:**

1. Overall, the article is difficult to read and follow, with significant omissions and formatting errors. For instance, Tables 2-4 lack the top and bottom rules, and the content for Appendices A.1-A.3 is missing. I do not believe the article is ready for publication in its current state.
2. I question the rationale behind posing ambiguous questions that are fundamentally flawed and should not be answered, rather than enhancing clarity by incorporating additional coordinate information.
3. While the authors introduce a novel dataset aimed at addressing ambiguity, it remains unclear whether this dataset is large and diverse enough to effectively evaluate the performance of various EQA models.
4.The selection of baseline models is limited. Although the authors acknowledge the challenges faced by modern EQA methods, they do not provide a comprehensive discussion of the specific limitations these models encounter when addressing ambiguous questions in 3D environments.

**Questions:**

See above

---

### Note · Authors · 2024-11-24

I have read and agree with the venue's withdrawal policy on behalf of myself and my co-authors.